# WARPFACE: REVISITING FACE REENACTMENT VIA SELF-SUPERVISED MOTION LEARNING IN DIFFUSION MODELS

## ABSTRACT

Face reenactment enables personalized motion transfer between identities and serves as a fundamental task in human animation. While recent diffusion-based approaches have achieved impressive visual quality, they often depend on human-centric inductive biases (e.g., landmark detectors), which limits their flexibility and scalability. In contrast, self-supervised GANs have demonstrated that meaningful motion representations can emerge directly from raw videos. This motivate us to introduce a novel self-supervised diffusion framework for face reenactment that eliminates the need for domain-specific priors. Our key insight is that diffusion models inherently encode rich motion cues, but naive extraction often leads to semantic collapse, where motion representations lose discriminability. To address this, we propose **WarpFace** with two core components: (1) Warping-enhanced Cross-Attention (*WarpCA*), which incorporates geometry-aware warping within the attention mechanism to enable robust motion learning while preventing semantic collapse; and (2) a Multi-Group Motion Encoder (*MGME*) that disentangles motion into structured subspaces for fine-grained control. Extensive experiments demonstrate that our method achieves expressive and accurate reenactment without relying on manual annotations or human-specific pretrained priors.

## 1 INTRODUCTION

Face reenactment, the task of transferring facial dynamics from a driving video to a source identity, has garnered significant attention due to its broad applications in entertainment, digital agents, and virtual communication. Despite the diversity of generative paradigms, face reenactment methods fundamentally rely on two core components, as illustrated in Table 1: (1) **driving condition**, which encodes the motion signal, and (2) **driving function**, which defines how this signal is applied to generate the desired animation.

Table 1: Face reenactment across generative paradigms: Driving Conditions vs. Driving Functions. CA denotes Cross-Attention mechanism for brevity.

| Cond \ Func | GAN | | Diffusion | | |
|---|---|---|---|---|---|
| | Warping | CA | Warping | CA | CA + Warping |
| Explicit Keypoint | ✓ | ✗ | ✗ | ✓ | ✗ |
| Implicit Keypoint | ✓ | ✗ | ✗ | ✗ | ✗ |
| Latent Vector | ✓ | ✗ | ✗ | ✓ | ✓(**Ours**) |

Recently, diffusion models (Ho et al., 2020) have emerged as the dominant paradigm for face reenactment, achieving superior visual fidelity through iterative denoising. However, existing methods (Ma et al., 2024; Xie et al., 2024) heavily rely on off-the-shelf facial landmarks as the driving condition, introducing strong human-specific pretrained priors into the learning process. While effective, this reliance limits the potential to scale further in a fully data-driven and flexible manner. Efforts to relax this dependency remain limited. For instance, X-NEMO (Zhao et al., 2025) utilizes an auxiliary facial appearance model, while SeMo (Zhang et al., 2025) employs noise masking strategies that struggle to isolate motion with fine-grained control.

In contrast, GAN-based methods have shown a distinct paradigm, enabling purely self-supervised training independent of third-party networks that supply facial priors. They enforce geometry-invariance on implicit keypoints (Siarohin et al., 2019; Wang et al., 2021) and apply information

bottlenecks to disentangle latent motion (Bounareli et al., 2023; Wang et al., 2024). Despite these advantages, they still face challenges in achieving stable training and high-fidelity animation.

The above analysis reveals a fundamental gap: diffusion methods achieve better visual fidelity but heavily rely on human-specific priors, while GAN methods learn meaningful motion independently in a self-supervised way but suffer from inferior quality and training instability. This motivates our research question: *Can diffusion models enable self-supervised motion disentanglement as GAN methods, while maintaining high-quality animation?*

**Our Key Insight.** We first investigate whether diffusion models inherently possess motion disentanglement capabilities. AnimateAnyone (Hu, 2024) has shown that identity can be preserved via mutual self-attention (MSA), eliminating the need for external appearance priors. Building on this, we further remove the cross-attention module that injects facial keypoints (Appx. B for experiment details). Interestingly, without explicit driving signal, the model produces random yet diverse expressions at inference, unveiling an *implicit motion space* embedded within the diffusion architecture (Figure 1, top).

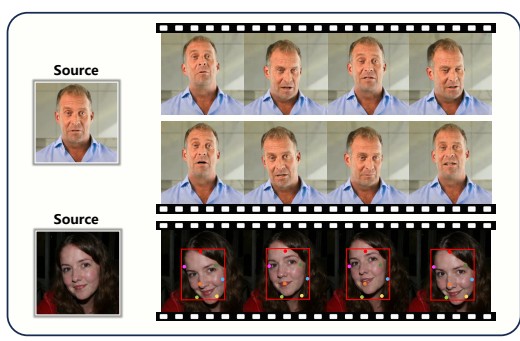

Figure 1: **Our Key Insight. Top**: a diffusion model trained solely on raw videos captures rich facial motions, revealing its intrinsic motion space. **Bottom**: a naive diffusion model variant using the geometric equivalence loss (Siarohin et al., 2019) for self-supervised keypoint extraction and animation, suffers from semantic collapse, with keypoints collapsing to nearly fixed positions. See **supplementary video** for clearer visualization.

Motivated by this discovery, we next attempt to exploit this latent motion space by integrating a self-supervised keypoint extractor (Wang et al., 2021) under geometry-invariance constraints, coupled with cross-attention for conditional injection. However, this approach leads to **semantic collapse** (see Figure 1, bottom): the extracted keypoints converge to nearly fixed spatial positions across diverse expressions. Such collapse mainly stems from the fact that motion features need to remain compatible across multi-scale feature maps and varying signal-to-noise ratios throughout the UNet-based diffusion process; with a zero-initialized motion extractor and no explicit supervision, features naturally converge across examples and lose their perceptual discriminability. These observations suggest that the challenge is not the absence of motion representation in diffusion models, but the difficulty of effectively accessing and controlling this latent space.

To bridge this gap, we systematically analyze architectural differences between GAN and diffusion approaches. We identify a key distinction from Table 1: while diffusion methods rely on standard cross-attention for condition injection, state-of-the-art GAN methods employ geometry-aware *warping* operations that preserve spatial relationships during motion transfer. This insight leads to our first contribution: *WarpCA (Warping-enhanced Cross-Attention)*, which enhances cross-attention layers with timestep-aware warping capabilities to enable effective motion disentanglement in diffusion models. Furthermore, to address the expressiveness limitations of sparse keypoints (Zhao et al., 2021), we propose *MGME (Multi-Group Motion Encoder)*, which extracts multiple groups of motion vectors across distinct linear subspaces. This design preserves semantic composability while enhancing control over challenging expressions. In summary, our main contributions are:

- We introduce a novel diffusion-based face reenactment framework—**WarpFace**, which achieves controllable motion disentanglement without human-specific priors, bridging high-quality animation and self-supervised learning.

- We propose *WarpCA*, a simple geometry-aware motion injection module, enabling robust self-supervised motion learning in diffusion models without semantic collapse.

- We design *MGME*, a multi-subspace motion encoder that improves expressiveness over complex facial dynamics while maintaining semantic composability.

- Extensive experiments validate our design choices and demonstrate superior or competitive performance compared to existing methods across multiple datasets.

## 2 RELATED WORKS

### 2.1 FACE REENACTMENT

Early face reenactment methods predominantly leveraged GANs (Goodfellow et al., 2014) for motion transfer, categorized by their motion representations: explicit methods use structured information like 3D parameters or landmarks extracted by off-the-shelf detectors (Kim et al., 2018; Nirkin et al., 2019; Ren et al., 2021; Doukas et al., 2021; Yin et al., 2022), while implicit methods embed motion as implicit keypoints or vectors in latent spaces (Siarohin et al., 2019; Wang et al., 2021; Drobyshev et al., 2022; Pang et al., 2023; Wang et al., 2024; Burkov et al., 2020; Guo et al., 2024a). Notable works include FOMM (Siarohin et al., 2019), which achieves motion disentanglement via the geometric invariance of keypoints, and LIA (Wang et al., 2024), which uses information bottlenecks for latent motion vectors. However, GAN-based generators remain limited in handling extreme expressions and diverse portrait styles.

Recent diffusion models (Ho et al., 2020) have demonstrated superior generation quality, with portrait animation methods adapting pre-trained LDMs through various conditioning strategies (Hu, 2024; Xie et al., 2024; Yang et al., 2024; Zhu et al., 2024; Ma et al., 2024). These approaches typically adopt siamese architectures with mutual self-attention (Vaswani et al., 2017) and temporal modules (Guo et al., 2024b), but rely heavily on explicit motion representations—facial keypoints and mesh renderings. Synthetic cross-identity data via ControlNet (Zhang et al., 2023) and auxiliary networks are also explored for motion control. Recent self-supervised attempts like SeMo (Zhang et al., 2025) struggle with controllability among latent motion vectors, as learned representations lack semantic constraints to ensure meaningful motion disentanglement.

Unlike these approaches, our **WarpFace** integrates geometry-aware warping directly into diffusion architectures, achieving controllable motion disentanglement without explicit supervision while maintaining superior quality.

### 2.2 REPRESENTATIONS FROM DIFFUSION MODELS

Recent investigations have shown that diffusion models (Ho et al., 2020) learn rich semantic representations beyond generation. Pioneering works by GD (Mukhopadhyay et al., 2023) and I-DAE (Chen et al., 2024) revealed inherent discriminative properties within diffusion features, catalyzing successful adaptations for diverse downstream tasks including correspondence estimation and object detection (Hedlin et al., 2023; 2024; Chen et al., 2023). These representations have also proven effective for generative applications such as image editing. Notably, MasaCtrl (Cao et al., 2023) and ConsiStory (Tewel et al., 2024) employ attention-based mechanisms to enable precise appearance transfer and consistent generation with structural coherence.

In contrast, **WarpFace** contributes to this field by demonstrating how geometry-aware warping can effectively harness diffusion representations for controllable facial motion transfer.

## 3 METHODOLOGY

### 3.1 OVERVIEW

Given a source portrait $I_s$, our goal is to generate a face reenactment sequence $\{I_{s \to d_i}\}_{i=1}^{L}$ conditioned on a driving video $\{I_{d_i}\}_{i=1}^{L}$ (or a single image $I_d$, when $L = 1$). Although training is constrained to same-identity reconstruction due to the scarcity of cross-identity pairs with matching expressions, the model can still achieve effective motion transfer across different identities at inference. This capability comes from learning an identity-agnostic motion representation without sacrificing facial expressiveness.

Our approach achieves high-fidelity face reenactment through latent diffusion models without relying on extra human-related priors such as 3DMM (Blanz & Vetter, 1999) or facial landmarks (Yang et al., 2023). Instead, we leverage the powerful generative capability of pre-trained diffusion models, specifically Stable Diffusion v1.5 (Rombach et al., 2022), to learn motion representations directly from data in a self-supervised manner. The overall architecture is illustrated in Figure 2.

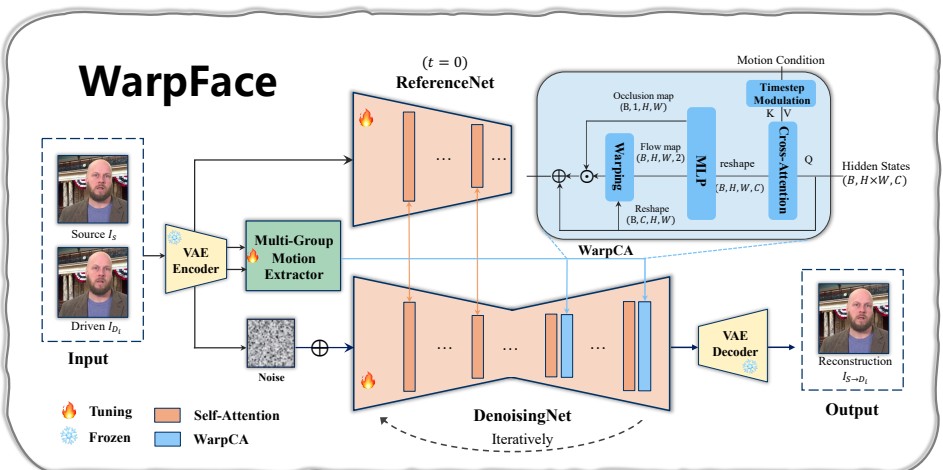

Figure 2: **Overall Architecture of WarpFace.** Our method enables controllable facial animation using diffusion models without relying on human-specific pretrained priors. The key innovation is a *WarpCA* (Warp-guided Cross-Attention, top-right) structure that performs geometry-aware feature warping within the attention mechanism. $\oplus$ denotes element-wise addition, and $\odot$ represents Hadamard product.

**WarpFace** builds on the dual-UNet latent diffusion paradigm and comprises three key components operating in VAE latent space: (1) to achieve robust identity preservation, we adopt a *ReferenceNet* that shares multi-scale features with the DenoisingNet via mutual self-attention, following established methods (Hu, 2024; Tian et al., 2024); (2) a *Multi-Group Motion Encoder (MGME)* that decomposes facial movements into orthogonal subspaces, enabling semantically meaningful motion representation; and (3) a *DenoisingNet* equipped with the proposed *WarpCA* that applies geometry-aware transformations to prevent semantic collapse and facilitate self-supervised motion learning.

## 3.2 PRELIMINARY

### 3.2.1 LATENT DIFFUSION MODELS

Latent Diffusion Models (LDMs) (Rombach et al., 2022) operate in the latent space of a pre-trained autoencoder $\mathcal{E}$, mapping images $x \in \mathbb{R}^{H \times W \times 3}$ to latent representations $z = \mathcal{E}(x) \in \mathbb{R}^{h \times w \times c}$, where $h = H/f$, $w = W/f$, and $f$ is the downsampling factor. The forward diffusion process gradually adds Gaussian noise:

$$q(z_t|z_0) = \mathcal{N}(z_t; \sqrt{\bar{\alpha}_t}z_0, (1 - \bar{\alpha}_t)\mathbf{I}), \tag{1}$$

where $\bar{\alpha}_t = \prod_{s=1}^{t} \alpha_s$ with $\alpha_t = 1 - \beta_t$ (noise schedule). The reverse process learns to predict noise $\epsilon$ through:

$$\mathcal{L}_{LDM} = \mathbb{E}_{z_0, \epsilon \sim \mathcal{N}(0,1), t} \left[ ||\epsilon - \epsilon_\theta(z_t, t, c)||_2^2 \right], \tag{2}$$

where $c$ represents conditioning information and $\epsilon_\theta$ is the denoising network.

### 3.2.2 LINEAR MOTION REPRESENTATION

We adopt a linear motion decomposition framework (Wang et al., 2024) where facial poses are represented in a structured latent space $\{p_j\}_{j=1}^n$ with orthogonal basis vectors $p_j$ encoding fundamental motion components. In this framework, any facial pose $\boldsymbol{\theta}_s$ decomposes relative to a globally canonical reference $\boldsymbol{\theta}_r$ as:

$$\boldsymbol{\theta}_s = \boldsymbol{\theta}_r + \boldsymbol{\theta}_{r \to s}, \quad \boldsymbol{\theta}_{r \to s} = \sum_{j=1}^{n} \alpha_j p_j, \tag{3}$$

where $\{\alpha_j\}$ are motion coefficients. Finally, motion transfer between source pose $\boldsymbol{\theta_s}$ and driving pose $\boldsymbol{\theta_d}$ exploits the additivity property:

$$\boldsymbol{\theta}_{s \to d} = \boldsymbol{\theta}_{r \to d} - \boldsymbol{\theta}_{r \to s}. \tag{4}$$

Consequently, this formulation enables identity-agnostic motion transfer through linear operations in motion space, ensuring both interpretability and controllability.

### 3.3 WARPFACE

#### 3.3.1 MULTI-GROUP MOTION ENCODER (MGME)

Conventional motion encoders represent facial dynamics as single vectors or keypoints, limiting expressiveness for hierarchical facial movements spanning different semantic regions. We propose *Multi-Group Motion Encoder (MGME)* that decomposes facial motion into $K$ orthogonal semantic subspaces for motion representation. As illustrated in Figure 3, *MGME* operates through two key steps:

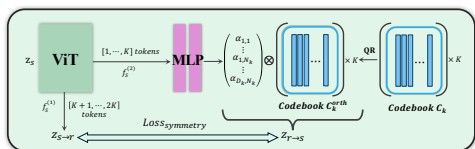

Figure 3: **Details for Multiple-Group Motion Encoder (MGME)**. The symbol $\otimes$ represents linear combination of vectors, and **QR** denotes QR decomposition for codebook orthogonalization.

*Hierarchical Feature Decomposition.* Given a source image $I_s$, we obtain the VAE latent $z_s = \mathcal{E}(I_s)$ and extract two features $\mathbf{f}_s^{(1)}, \mathbf{f}_s^{(2)}$ with a zero-initialized ViT (Dosovitskiy et al., 2021). These two features respectively support reverse and forward motions, working with symmetry-based regularization via Eq. 13 to better exploit the latent information.

$$\mathbf{f}_s^{(1)}, \mathbf{f}_s^{(2)} = \mathcal{F}_{ViT}(z_s). \tag{5}$$

Both are split into $K$ groups $\mathbf{f}_{s,k}^{(1)}, \mathbf{f}_{s,k}^{(2)}$ and mapped by group-specific MLPs:

$$\boldsymbol{\theta}_{s \to r} = \{\text{MLP}_k^{(1)}(\mathbf{f}_{s,k}^{(1)})\}_{k=1}^K, \quad \boldsymbol{\alpha}_k = \text{MLP}_k^{(2)}(\mathbf{f}_{s,k}^{(2)}), \ k = 1, \dots, K, \tag{6}$$

where $\boldsymbol{\theta}_{s \to r}$ is the motion latent from source $\boldsymbol{\theta}_s$ to canonical pose $\boldsymbol{\theta}_r$, and $\{\boldsymbol{\alpha}_k\}$ provide coefficients for the reverse motion transformation $\boldsymbol{\theta}_{r \to s}$.

*Linear Motion Combination.* For each group, we maintain a learnable codebook $\mathbf{C}_k \in \mathbb{R}^{D_k \times N_k}$, where $D_k$ and $N_k$ denote the feature dimension and the number of basis vectors within each group. Orthogonality is enforced through QR decomposition to promote semantic disentanglement. The group-wise motion representation is computed as:

$$\mathbf{g}_k = \sum_{n=1}^{N_k} \alpha_{k,n} \mathbf{c}_{k,n}^{\text{orth}}, \quad \text{where } \mathbf{C}_k^{\text{orth}} = \text{QR}(\mathbf{C}_k). \tag{7}$$

For the global motion representation, we combine all semantic groups:

$$\boldsymbol{\theta}_{r \to s} = \text{Concat}[\mathbf{g}_1, \mathbf{g}_2, \dots, \mathbf{g}_K]. \tag{8}$$

The same procedure is applied to the driving image $I_d$, yielding $\boldsymbol{\theta}_{r \to d}$, which is further used to compute the final motion transfer condition $\boldsymbol{\theta}_{s \to d}$ via Eq. 4.

This overall design not only enables fine-grained motion control but also maintains linear additivity for cross-identity transfer through orthogonal disentanglement, thereby facilitating interpretable and controllable face animation with enhanced expressiveness.

#### 3.3.2 WARPING-ENHANCED CROSS-ATTENTION (WARPCA)

The naive cross-attention mechanism in DenoisingNet suffers from feature degradation as it repeatedly aggregates identical motion features across multiple layers and denoising timesteps. This design requires motion features to remain compatible across multi-scale feature maps and varying signal-to-noise ratios, inevitably causing them to lose discriminative power during the aggregation process. We address this problem by introducing geometric correspondence into cross-attention layers using differentiable flow-based warping operations.

As illustrated in Figure 2 (top-right), *WarpCA* operates through the following sequential steps. First, we embed the latent motion conditions into time-aware representations. Given a motion latent $\boldsymbol{\theta}_{s \to d} \in \mathbb{R}^{D_m}$ and a timestep $t \in \{1, 2, \dots, T\}$, the time-aware modulation is defined as:

$$\boldsymbol{\theta}_{s \to d}^{(t)} = \left(1 + \boldsymbol{\gamma}^{(t)}\right) \times \boldsymbol{\theta}_{s \to d} + \boldsymbol{\delta}^{(t)} \tag{9}$$

where $\boldsymbol{\gamma}^{(t)}$ and $\boldsymbol{\delta}^{(t)} \in \mathbb{R}^{D_m}$ are obtained from the timestep embedding via separate linear layers.

Then, we perform geometry-guided attention computation. The modulated representation $\boldsymbol{\theta}_{s \to d}^{(t)}$ serves as the *key* and *value* ($\mathbf{K}, \mathbf{V}$) in cross-attention, where the *query* $\mathbf{Q} \in \mathbb{R}^{B \times (H \times W) \times C}$ comes from the hidden states of the previous layer. The attention output is reshaped to a spatial layout $\mathbb{R}^{B \times H \times W \times C}$ and passed through two separate MLPs to predict:

- A **flow map** $\mathbf{F} \in \mathbb{R}^{B \times H \times W \times 2}$, capturing per-pixel displacement vectors that encode geometric correspondence—i.e., where each position in the driven image $I_d$ should be sampled from the source image.
- An **occlusion map** $\mathbf{O} \in [0,1]^{B \times H \times W \times 1}$, modeling per-pixel visibility confidence to handle occlusions. It shows whether each position can be faithfully reconstructed from the source image by warping operation.

After that, we perform occlusion-aware feature fusion. Specifically, the flow map $\mathbf{F}$ is applied to warp the original hidden state $\mathbf{H} \in \mathbb{R}^{B \times H \times W \times C}$, producing motion-aligned features $\mathbf{H}_{\text{warp}}$. We then apply residual fusion weighted by the occlusion map:

$$\hat{\mathbf{H}} = \mathbf{H} + \mathbf{O} \odot \mathcal{W}(\mathbf{H}, \mathbf{F}), \tag{10}$$

where $\mathcal{W}(\cdot, \cdot)$ denotes the differentiable warping operation and $\odot$ denotes the Hadamard product.

Unlike naive cross-attention prone to semantic collapse due to repeated feature aggregation, *WarpCA* explicitly models geometric transformations and timestep information to maintain motion discriminability with lightweight network modification and minimal computational overhead.

### 3.3.3 TRAINING OBJECTIVE

Our training objective combines adaptive reconstruction loss with motion symmetry regularization:

$$\mathcal{L}_{\text{total}} = w_{\text{adapt}} \cdot \mathcal{L}_{\text{LDM}} + \lambda_{\text{sym}} \cdot \mathcal{L}_{\text{sym}}. \tag{11}$$

To prioritize learning on regions with significant motion, we introduce an adaptive weighting mechanism that emphasizes high-variation areas:

$$w_{\text{adapt}}(i,j) = \begin{cases} \tau, & \text{if } \delta_{i,j} \leq \bar{\Delta} \\ \tau \cdot (1 + \frac{\delta_{i,j}}{\bar{\Delta}} \cdot \epsilon), & \text{otherwise} \end{cases} \tag{12}$$

where $\delta_{i,j} = |z_s(i,j) - z_d(i,j)|$ and $\bar{\Delta} = \frac{1}{HW} \sum_{i,j} |z_s(i,j) - z_d(i,j)|$ respectively represent pixel-wise and mean latent differences between paired images, while $H$ and $W$ are corresponding height and width of the latent feature maps.

To ensure symmetry property of motion transformations, we enforce bidirectional consistency through:

$$\mathcal{L}_{\text{sym}} = \|\boldsymbol{\theta}_{s \rightarrow r} + \boldsymbol{\theta}_{r \rightarrow s}\|_2^2. \tag{13}$$

The hyperparameters $\lambda_{\text{sym}}$, $\tau$, and $\epsilon$ control the relative importance of symmetry regularization and adaptive weighting strength, respectively.

## 4 EXPERIMENTS

### 4.1 EXPERIMENTAL SETUP

**Datasets.** We train **WarpFace** on three public datasets: HDTF (Zhang et al., 2021), VFHQ (Xie et al., 2022), and Hallo3 (Cui et al., 2025). All videos are first segmented into clips of maximum 10 seconds, where source-driven image pairs are randomly selected if the interval is larger than 0.2s. All clips are resampled to 25fps and center-cropped to 256 resolution. We adopt augmentations like PC-AVS (Zhou et al., 2021) for *MGME* to enhance identity-independent representation learning. For evaluation, we use the VFHQ official test set (50 videos) and randomly sample another 50 videos from HDTF and Hallo3, totaling 100 videos.

**Implementation Details.** We implement **WarpFace** using PyTorch and train on 4 NVIDIA A6000 GPUs for 50K steps with a total batch size of 72. We use AdamW (Loshchilov & Hutter, 2017) optimizer with a learning rate 2e-5. We employ 1000 denoising steps for diffusion training, and DDIM sampling (40 steps) with classifier-free guidance (scale=2.5) during inference. For DenoisingNet, motion vectors are injected into *middle and upsampling layers* via *WarpCA*, while appearance features are integrated into *downsampling and middle layers* through mutual self-attention. For *MGME*, we have $K = 4$ motion subspaces, each with a codebook of $N_k = 128$ vectors at $D_k = 256$ dimensions. The loss hyperparameters are $\lambda_{\text{sym}} = 0.1$, $\tau = 0.5$, $\epsilon = 2.0$ respectively.

**Evaluation Metrics.** We evaluate identity preservation using cosine similarity (CSIM) of Arc-Face (Deng et al., 2019) features, reconstruction and image quality via LPIPS (Johnson et al., 2016)

Table 2: **Quantitative comparison on self-reenactment and cross-reenactment.** Methods are grouped as GAN-based (top) and diffusion-based (bottom); best results are in **bold**, second-best are underlined; User Pref. denotes user top preference in percentage; * denotes our reproduced version while '−' means not applicable due to pronounced artifacts.

| Method | Self-reenactment | | | | | Cross-reenactment | | | |
|---|---|---|---|---|---|---|---|---|---|
| | CSIM ↑ | APD ↓ | AED ↓ | LPIPS ↓ | FID ↓ | CSIM ↑ | APD ↓ | AED ↓ | User Pref(%). ↑ |
| FOMM | 0.52 | 8.91 | 15.64 | 0.44 | 44.1 | 0.37 | 14.83 | 22.72 | − |
| LIA | 0.60 | 9.20 | 14.32 | 0.37 | 40.4 | 0.40 | 14.66 | 22.15 | − |
| HyperReenact | 0.59 | 6.73 | 12.17 | 0.32 | 35.6 | 0.41 | 11.06 | 18.45 | − |
| LivePortrait | 0.78 | **2.24** | **5.44** | **0.19** | 15.7 | **0.65** | **5.97** | 10.41 | 36.8 |
| DiffusionAct | 0.64 | 6.47 | 12.09 | 0.31 | 25.2 | 0.43 | 11.01 | 17.87 | − |
| SeMo* | 0.65 | 4.52 | 10.23 | 0.29 | 18.5 | 0.46 | 9.76 | 15.14 | 11.4 |
| X-NEMO* | 0.72 | 5.36 | 8.92 | 0.26 | 19.8 | 0.57 | 8.13 | 13.91 | 12.5 |
| **Ours** | **0.79** | 2.51 | 5.58 | 0.21 | **9.6** | **0.65** | 6.10 | **10.28** | **40.3** |

and FID (Heusel et al., 2017) respectively. For motion transfer evaluation, we compute Average Pose Distance (APD) and Average Expression Distance (AED) following (Ren et al., 2021). In terms of self-reenactment, all metrics are compared between driven and generated images. For cross-reenactment, CSIM and is computed by source and generated images, while APD and AED measure motion transfer accuracy between driven and generated ones.

**Baselines.** Methods with publicly available training code were retrained on all three datasets; those with only pretrained weights and inference code were evaluated directly. Fully non-public methods were re-implemented following their original protocols.

We compare four GAN-based methods: FOMM and LivePortrait (Siarohin et al., 2019; Guo et al., 2024a) (implicit keypoints, with LivePortrait evaluated in inference-only mode), and LIA and HyperReenact (Wang et al., 2024; Bounareli et al., 2023) (latent motion). For diffusion-based baselines, we consider landmark-driven DiffusionAct (Bounareli et al., 2025) as well as implicit motion latent-driven methods. Among them, X-NEMO (Zhao et al., 2025) and SeMo (Zhang et al., 2025), are closest to our self-supervised denoising setting. We re-implemented both, but due to the absence of large internal high-quality datasets and the sensitivity of unstable training strategies (e.g., GANs and masking), their reproduced performance falls below the original reports.

### 4.2 COMPARISON

**Quantitative Results.** Table 2 presents comprehensive quantitative results on both self- and cross-reenactment. Our **WarpFace** demonstrates superior or competitive performance across most metrics, achieving state-of-the-art results in identity preservation and image quality.

*Self-reenactment.* For each video, the first frame is used as the source, and 20 eligible frames (>0.2s apart) are randomly sampled to form 20 source–target pairs. **WarpFace** achieves the highest identity similarity and best image quality, significantly outperforming both GAN-based and diffusion-based methods. While LivePortrait excels in pose transfer accuracy and perceptual quality, **WarpFace** maintains competitive performance with slightly better identity preservation. Compared to diffusion-based approaches, **WarpFace** surpasses both explicit keypoint and implicit motion vector based approaches across all metrics, demonstrating the effectiveness of our overall self-supervised diffusion framework.

*Cross-reenactment.* We randomly split the 100 test videos into 50 source-driven video pairs, using the first frame from source video as identity reference and 20 randomly sampled frames from driven video as targets. Our **WarpFace** achieves comparable identity preservation to LivePortrait while maintaining competitive pose transfer accuracy. We further conduct a user study following (Bounareli et al., 2023), presenting 20 image pairs to 30 volunteers for overall evaluation on identity preservation, pose transfer, and image quality. Notably, methods with FID > 25 are excluded in advance, given their pronounced visual artifacts. **WarpFace** achieves the highest user top-preference score of 40.3%, outperforming LivePortrait (36.8%) and significantly surpassing other baselines. These results demonstrate **WarpFace**'s strong generalization capability across different identities while maintaining high-quality motion transfer.

**Qualitative Results.** As mentioned earlier, we discard methods that produce evident visual flaws with FID greater than 25. Figure 4 and more results in Appendix F present comprehensive visual

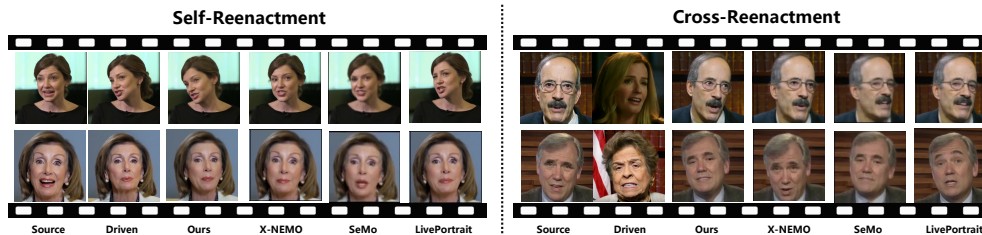

**Self-Reenactment**

Source Driven Ours X-NEMO SeMo LivePortrait

**Cross-Reenactment**

Source Driven Ours X-NEMO SeMo LivePortrait

Figure 4: **Qualitative Comparisons.** Left: self-reenactment; right: cross-reenactment. To better distinguish between methods, we recommend comparing the Driven and Generated images in terms of head pose, expression, and eye gaze. **WarpFace** achieves accurate motion transfer and consistent identity preservation.

comparisons. Frequently, competing approaches suffer from unfaithful motion transfer. However, **WarpFace** demonstrates strong capability in generating high-fidelity outputs that preserve source appearance while accurately transferring target facial dynamics, including extreme facial expressions, head poses and subtle eye gaze directions.

### 4.3 ABLATION STUDIES

We conduct comprehensive ablation studies under the same self-reenactment evaluation protocol to validate our design choices across identity preservation (CSIM), motion accuracy (APD/AED), and image quality (FID). See the **Appendix** for more ablations(Appx. C) and visualizations(Appx. E).

**Component Effectiveness.** Table 3 presents the effectiveness of our core components. Comparing rows 1 vs. 2 (also, rows 3 vs. 5), our multi-group motion vectors consistently improve all metrics compared to implicit keypoints, validating the enhanced motion expressiveness achieved through structured representation. Comparison of rows 1 and 3 (also, 2 and 5) reveals clear improvements in motion transfer with *WarpCA*, confirming its benefit for motion disentanglement. The last two rows indicate that timestep modulation improves overall quality, with Scale and Shift trends in Figure 6 further validating its effect in denoising process. The combination of all components achieves optimal performance across all metrics.

Table 3: **Ablation study on the impact of different technical components.** (a) CA: standard cross-attention; (b) IP: implicit keypoint; (c) *WarpCA*: proposed warping-enhanced cross-attention; (d) GVector: multi-group latent motion vector; and (e) w/o timestep: *WarpCA* without timestep modulation. Our final approach uses *WarpCA* + GVector. **Best** results are in bold, and second-best are underlined.

| Index | Method | CSIM ↑ | APD/AED ↓ | FID ↓ |
|---|---|---|---|---|
| 1 | CA + IP | 0.71 | 6.84/13.15 | 23.7 |
| 2 | CA + GVector | 0.76 | 6.45/12.52 | 21.4 |
| 3 | *WarpCA* + IP | 0.72 | 4.09/9.04 | 17.6 |
| 4 | Ours w/o timestep | 0.77 | 2.88/6.07 | 12.5 |
| 5 | **Ours** | **0.79** | **2.51/5.58** | **9.6** |

Table 4: **Ablation study on the proposed design choices.** (a) **MSA** and (b) *WarpCA* represent mutual self-attention and cross-attention layers at different DenoisingNet depths (**down**sample/ **mid**dle/ **up**sample/ **all** layers), respectively; (c) Motion **Codebook** configurations with $K \times N_k$, while $D_k$ is fixed as 256. **Final selected values** are shown in red, **best** results are in bold, and second-best are underlined.

(a) MSA

|  | CSIM ↑ | APD/AED ↓ | FID ↓ |
|---|---|---|---|
| down | 0.75 | 2.69/5.68 | 10.2 |
| up | 0.70 | 4.12/9.27 | 11.3 |
| **down+mid** | 0.79 | **2.51/5.58** | 9.6 |
| mid+up | 0.71 | 3.74/8.15 | 10.8 |
| all | **0.84** | 2.91/6.24 | **8.9** |

(b) *WarpCA*

|  | CSIM ↑ | APD/AED ↓ | FID ↓ |
|---|---|---|---|
| down | 0.59 | 5.38/11.12 | 14.2 |
| up | 0.74 | 2.91/6.09 | 10.4 |
| down+mid | 0.64 | 4.97/10.26 | 13.5 |
| **mid+up** | **0.79** | 2.51/5.58 | **9.6** |
| all | 0.70 | **2.43/5.39** | 11.3 |

(c) Codebook

|  | CSIM ↑ | APD/AED ↓ | FID ↓ |
|---|---|---|---|
| 8 × 64 | 0.76 | 2.97/6.31 | 12.7 |
| **4 × 128** | **0.79** | **2.51/5.58** | **9.6** |
| 2 × 256 | 0.77 | 2.85/5.96 | 11.6 |
| 1 × 512 | 0.77 | 2.68/5.75 | 10.5 |
| 2 × 128 | 0.77 | 2.74/5.81 | 10.8 |
| 8 × 128 | 0.75 | 2.80/5.88 | 11.2 |

**Architecture Design.** Table 4 analyzes different attention layer configurations in **WarpFace** framework. For **MSA** layers that inject identity information, **downsampling** layers contribute more effectively than upsampling layers while maintaining motion transfer performance. Adding middle layers further enhances all metrics. Although utilizing all layers achieves the best identity preservation, the inclusion of upsampling layers impairs motion transfer accuracy, making the down+mid configuration the optimal choice. For *WarpCA* layers that integrate motion information, **upsampling** layers outperform downsampling layers without compromising identity preservation. While

using all layers achieves the best motion transfer performance, downsampling layers interfere with identity preservation, making mid+up our preferred configuration.

**Motion Codebook Configuration.** Starting from LIA's baseline configuration of 512 total vectors, we first explore subspace partitioning by dividing the codebook into 2/4/8 groups, keeping the same overall vector count. The results demonstrate that 4 subspaces achieve optimal performance across all evaluation metrics. We then investigate the impact of total codebook size by varying the number of groups, finding that the $4 \times 128$ configuration provides the best balance between motion expressiveness and computational efficiency. Excessive parameters result in underutilized codebook entries, while insufficient capacity limits the expressiveness of motion dynamics.

### 4.4 MOTION SPACE ANALYSIS

To validate the interpretability of multi-group motion representation, we analyze both subspace- and vector-level edits for real-world facial animation tasks.

**Subspace-Level Semantics.** We assess semantic alignment between motion subspaces and facial regions by varying the all coefficients of a single subspace. As shown in Figure 5 (left), adjusting weights of different subspaces $\mathcal{G}_k$ yields distinct changes: $\mathcal{G}_1$ affects dynamics around the eye region while $\mathcal{G}_2$ primarily controls various expression types.

**Vector-Level Interpretability.** For finer-grained analysis, we modify the coefficients for individual motion vectors within different subspaces. Figure 5 (right) shows each vector captures specific motion patterns: vectors in $\mathcal{G}_1 V_{37}$ encode vertical eye gaze states, while $\mathcal{G}_3 V_{64}$ vectors represent various head poses.

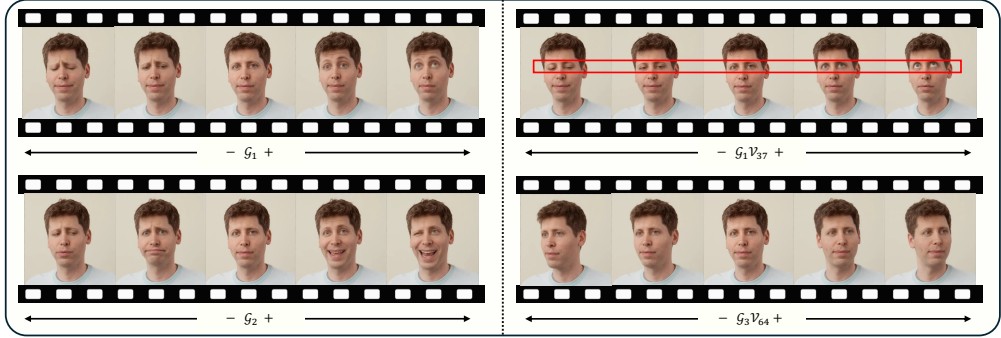

Figure 5: **Motion Space Exploration.** Manipulated results on *an in-the-wild example* show that subspace $\mathcal{G}_1$ corresponds to eye-region dynamics, while $\mathcal{G}_2$ captures diverse expression types. Here, $\mathcal{G}$ denotes a motion group, and $V$ is the vector index within a group. For instance, $\mathcal{G}_1 V_{37}$ and $\mathcal{G}_3 V_{64}$ are associated with vertical eye gaze and head shaking, respectively.

## 5 CONCLUSION AND LIMITATIONS

In this work, we have proposed a novel self-supervised diffusion framework for controllable motion disentanglement without human-specific priors. Our key insight reveals that diffusion models possess intrinsic motion spaces, but naive extraction leads to semantic collapse when motion representations degrade into non-discriminative features that fail to capture facial dynamics. We have addressed this with geometry-aware attention and multi-group motion decomposition, enabling effective self-supervised motion disentanglement.

Despite promising results, our **WarpFace** has several limitations. First, iterative denoising sampling hinders potential real-time applications. Second, we lack scalability validation due to computational constraints. Lastly, we observed that our approach tended to fail under extreme poses or heavy occlusion (as shown in Figure 9). Future works could explore distillation techniques for acceleration and investigate scalability with larger datasets to handle extreme cases. Overall, we hope this work will inspire advances in self-supervised representation disentanglement and controllable human motion generation with diffusion models.

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

APPENDIX

## A  LLM USAGE STATEMENT

We employed a Large Language Model (LLM) as a general-purpose assistive tool during the preparation of this submission. Specifically, the LLM was used for (i) correcting grammar errors, (ii) refining and polishing the writing style, and (iii) providing assistance with code implementation details. All generated text and code outputs were carefully reviewed, verified, and, where necessary, modified by the authors to ensure accuracy and originality. The LLM did not contribute to the conceptualization, experimental design, or interpretation of results. The authors take full responsibility for all content presented in this paper.

## B  VIDEO DEMONSTRATION

We strongly encourage readers to watch the supplementary video for additional visualizations.

**Intrinsic Motion Space.** As demonstrated in Figure 1, we construct a baseline similar to AnimateAnyone (Hu, 2024) using their open-source implementation[1] with two key modifications: (1) removing some modules (cross-attention and temporal layers) and keypoint components (landmark extractors and pose guider), and (2) applying only our adaptive LDM loss (Eq. 11). Although we train directly on same-identity and different-expression pairs without explicit motion conditioning, this baseline generates sequences with preserved identity but random poses, supporting our key insight that the motion space is inherently embedded within diffusion models.

**Temporal Consistency.** Despite using a text-to-image model (Stable Diffusion v1.5) and its corresponding VAE (Rombach et al., 2022) as our foundation without additional temporal layers, our learned motion space exhibits compactness and continuity. When driven by video sequences, our **WarpFace** produces smooth frame-to-frame transitions without severe temporal discontinuities. This validates the compactness of our motion representation.

## C  HYPERPARAMETER SENSITIVITY ANALYSIS

We conduct comprehensive hyperparameter sensitivity studies to validate **WarpFace**'s robustness. Following our self-reenactment evaluation protocol, we focus on the trends in motion transfer accuracy using APD and AED.

Table 5: **Loss Function Weight Analysis. Best** results are in bold, second-best are underlined for each hyperparameter and final selected values are shown in red.

| Parameter | Metric | Parameter Values | | | |
|---|---|---|---|---|---|
| | | 0.01 | 0.05 | 0.1 | 0.2 |
| $\lambda_{\text{sym}}$ | APD↓ | 3.07 | 2.65 | **2.51** | 2.79 |
| | AED↓ | 7.11 | 6.08 | **5.58** | 6.16 |
| | | 0.1 | 0.25 | 0.5 | 1.0 |
| $\tau$ | APD↓ | 3.54 | 2.89 | **2.51** | 2.85 |
| | AED↓ | 7.72 | 7.25 | **5.58** | 7.44 |
| | | 0.5 | 1.0 | 2.0 | 3.0 |
| $\epsilon$ | APD↓ | 3.23 | 2.99 | **2.51** | 2.94 |
| | AED↓ | 6.82 | 6.34 | **5.58** | 6.47 |

**Loss Function Weight Analysis.** In table 5, we systematically vary the weights of different loss components in our total loss (Eq. 11) while keeping other parameters fixed. Finally, we select the optimal configuration of $\lambda_{\text{sym}} = 0.1$, $\tau = 0.5$, $\epsilon = 2.0$ that achieves superior motion transfer quality.

---

[1] https://github.com/MooreThreads/Moore-AnimateAnyone

*MGME* **Architecture Configuration.** We analyze the impact of ViT backbone depth, hidden state dimension, and codebook dimension on motion transfer quality. Table 6 presents results for 4-10 layer configurations. A 6-layer ViT achieves the best trade-off between performance and efficiency. While deeper networks (8 and 10 layers) yield marginal improvements, they incur significantly higher computational costs. Finally, we choose the optimal configuration of a 6-layer ViT, hidden dimension $d = 256$, and codebook dimension $k = 256$.

Table 6: *MGME* **Architecture Configuration.** Best results are in **bold**, second-best are underlined for each configuration and final selected values are shown in red.

| Configuration | Metric | Parameter Values | | | |
|---|---|---|---|---|---|
| | | 4 | 6 | 8 | 10 |
| **ViT depth** | APD↓ | 3.31 | 2.51 | 2.44 | **2.41** |
| | AED↓ | 7.17 | 5.58 | 5.52 | **5.50** |
| | | 64 | 128 | 256 | 512 |
| **hidden dim** | APD↓ | 3.14 | 2.67 | 2.51 | **2.49** |
| | AED↓ | 6.55 | 5.94 | **5.58** | **5.55** |
| | | 128 | 256 | 512 | 768 |
| **codebook dim** | APD↓ | 2.95 | **2.51** | 2.53 | 2.81 |
| | AED↓ | 6.12 | 5.57 | **5.58** | 5.96 |

Our analysis reveals consistent performance across reasonable hyperparameter ranges, demonstrating the robustness of **WarpFace**. Based on the evaluation of both accuracy and computational efficiency, we select the optimal configuration that ensures practical deployment efficiency while maintaining superior motion transfer quality.

## D  SCALE–SHIFT ANALYSIS ACROSS TIMESTEPS

Following the self-reenactment reconstruction setting in quantitative results, we analyze the behavior of our timestep-aware modulation mechanism in *WarpCA* by examining how the scale ($\gamma^{(t)}$) and shift ($\delta^{(t)}$) parameters evolve during the denoising process. Figure 6 shows the average values of these parameters across different timesteps, providing insights into how *WarpCA* adaptively modulates motion features throughout the diffusion process.

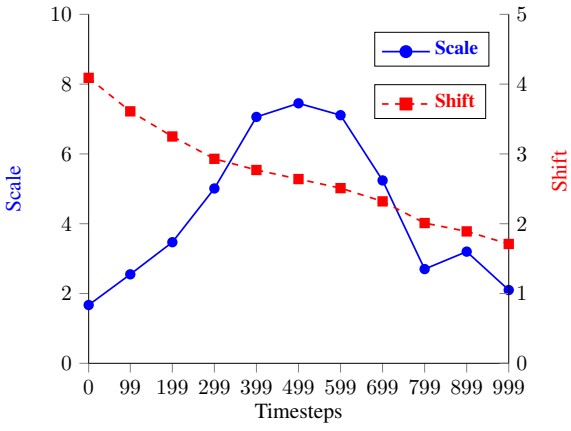

Figure 6: Timestep-wise modulation of Scale and Shift.

The analysis reveals a clear pattern: scale parameters remain relatively small at both initial and final denoising stages, while shift parameters gradually decrease throughout the process. This behavior aligns with established diffusion principles where denoising follows a coarse-to-fine reconstruction

paradigm. Specifically, our network follows a three-stage progression of identity recovery, pose refinement, and detail enhancement. The motion condition primarily operates in the middle stage, while the small scale values in the early and late stages reflect conservative modulation—supporting global structure establishment at the beginning and fine detail restoration at the end.

## E CANONICAL FACE VISUALIZATION

We visualize the learned canonical face representations to examine the structure of our motion space. To probe the semantics of the motion representation, we consider two conditions: (1) null motion ($\boldsymbol{\theta}_m = 0$) and (2) canonical-pose motion ($\boldsymbol{\theta}_m = \boldsymbol{\theta}_{s \to r}$), where $r$ denotes the canonical face for the current identity.

As shown in Figure 7, the null condition reconstructs the input with original identity and facial pose, while the canonical code consistently drives faces toward the reference pose without altering identity. This demonstrates that our motion space is semantically structured, with disentangled control over pose and appearance.

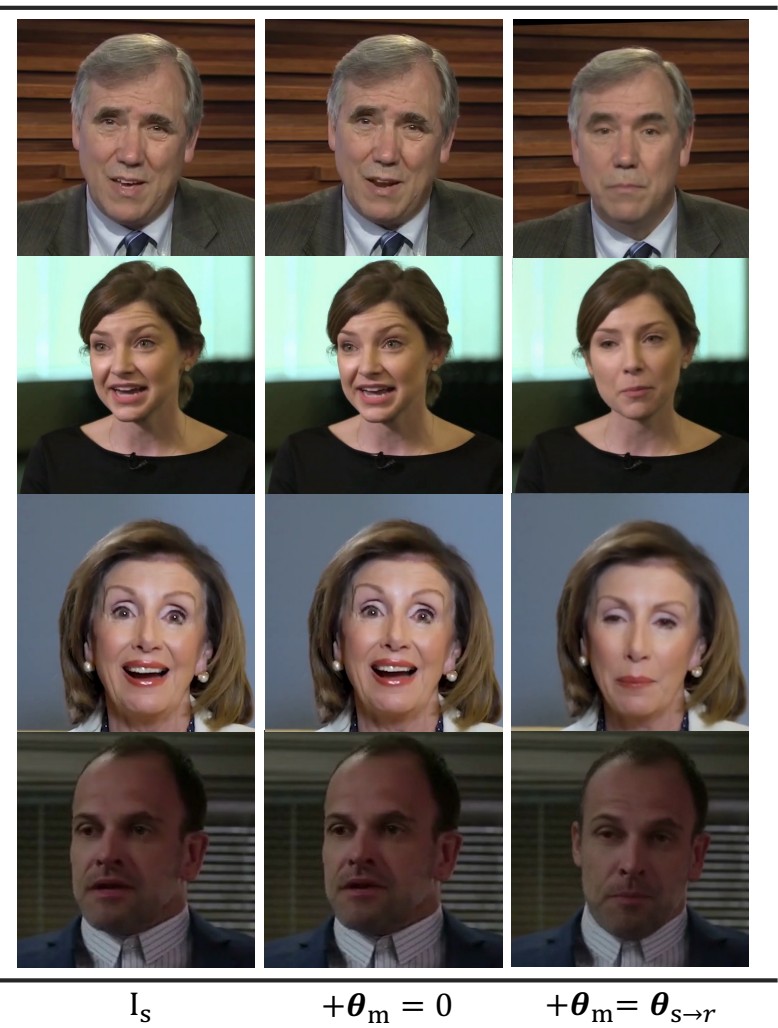

$$I_s \qquad +\boldsymbol{\theta}_m = 0 \qquad +\boldsymbol{\theta}_m = \boldsymbol{\theta}_{s \to r}$$

Figure 7: **Canonical Face Visualization.** Column 1 shows the source image. Column 2 depicts reconstructions with null motion ($\boldsymbol{\theta}_m{=}0$), revealing robust identity preservation. Column 3 applies canonical-pose conditioning ($\boldsymbol{\theta}_m{=}\boldsymbol{\theta}_{s \to r}$), which consistently drives diverse identities toward the reference pose without altering appearance.

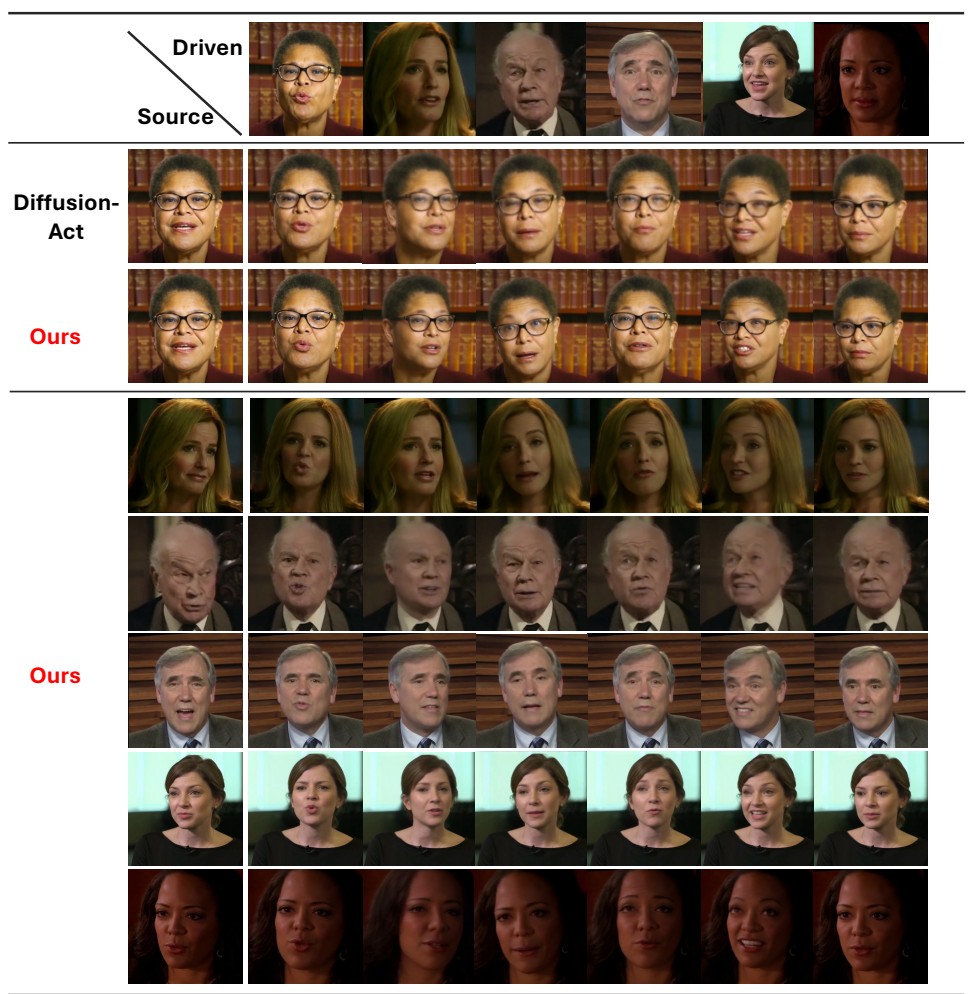

Figure 8: **Extended Visualization Results.** The first two rows compare our **WarpFace** with a diffusion-based baseline using external keypoints. Our results exhibit sharper appearance, while DiffusionAct (Bounareli et al., 2025) benefits from external keypoints, leading to slightly better alignment. Overall, both achieve comparable quality. Additional rows showcase **WarpFace**'s robustness across diverse scenarios, including cross-identity transfer, complex expressions, and challenging lighting. Also, we encourage row- and column-wise comparisons to better appreciate the consistency and generalization.

# F  ADDITIONAL QUALITATIVE RESULTS

**Extended Visualization Results.** Figure 8 (bottom) demonstrates **WarpFace**'s robustness across diverse scenarios including cross-identity transfer, complex expressions and varying lighting conditions.

**Comparison with Diffusion-based Methods using External Keypoints.** We compare **WarpFace** with DiffusionAct (Bounareli et al., 2025), a recent diffusion-based face reenactment method that utilizes external facial landmarks as driving conditions. The first two rows of Figure 8 show that **WarpFace** performs competitively without using landmarks, suggesting potential for further scalability without human-specific priors.

**Failure Case Analysis.** Figure 9 illustrates limitations of **WarpFace**: (1) extreme poses (close to 90° rotation) may cause appearance distortion due to invisible facial regions, and (2) partial occlusions may propagate to target poses. These cases highlight opportunities for future integration with 3D face modeling.

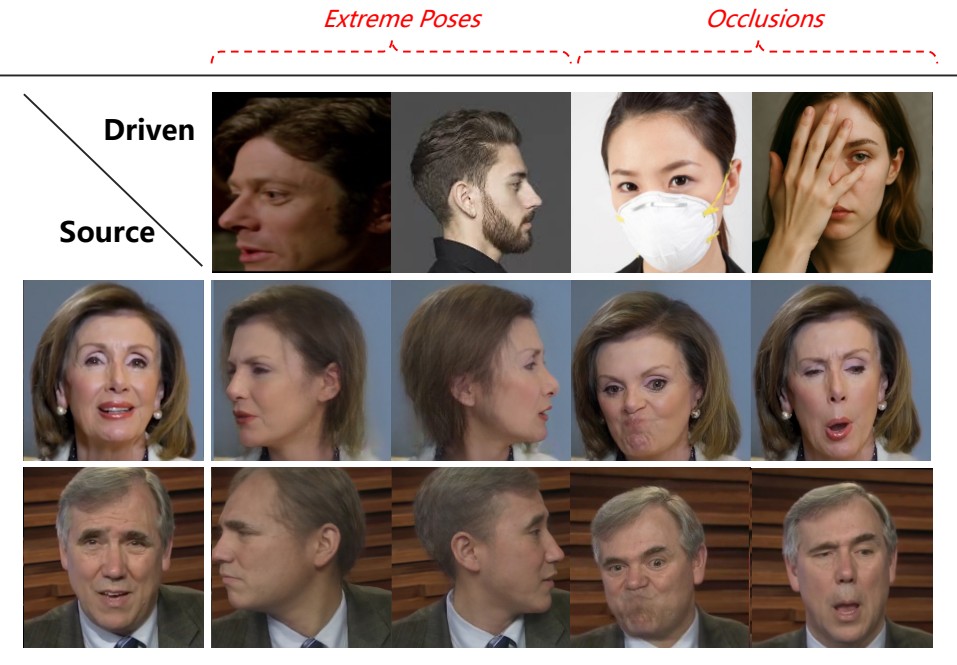

Figure 9: **Failure Case Analysis.** The first two columns show appearance distortion under extreme poses due to invisible facial regions. The last two columns illustrate failure cases under occlusion, where incorrect motion estimation leads to expression artifacts.

