# OpenReview forum: "WarpFace: Revisiting Face Reenactment via Self-Supervised Motion Learning in Diffusion Models"
_ICLR.cc/2026/Conference — Submitted to ICLR 2026_

### Official Review · Reviewer_PYDH · 2025-10-21

**Soundness:** 2
**Presentation:** 2
**Contribution:** 1
**Rating:** 2
**Confidence:** 3

**Summary:**

This paper proposes a self-supervised motion learning framework for face reenactment. To transfer the motion pattern without human annotations, this paper aims to exploit the rich motion cues encoded in the diffusion model. This paper proposes two components, including warping-enhanced cross-attention to capture geometry information and a multi-group motion encoder for fine-grained motion control. The experiments are conducted on self- and cross-reenactment setups.

**Strengths:**

1. Self-supervised learning for face reenactment has potential for better scaling the models.
2. The overall paper is easy to follow.

**Weaknesses:**

1. The experiment evaluation contains no comparisons with state-of-the-art video diffusion models (e.g., Wan2.1). As current video generative models become more and more versatile and powerful, it would be beneficial to compare the proposed method with the SoTA video foundation models to further support the contributions.
2. In Figure 2, the latent motion conditions are encoded from a single image. Given that the motion information is temporally dependent, how can this method derive the motion information simply from a static image?
3. In Figure 2, the source and driven images are sampled from the same identity. How to ensure the cross-identities transfer?
4. How to ensure the appearance and motion are properly disentangled? No objective ensures the appearance and motion patterns are decoupled. It might cause degraded results when the motion patterns transfer to different identities that are not similar to the source one.

**Questions:**

1. Can the proposed method be applicable to the facial images that are not precisely aligned in the middle of the image?

---

> ### Author Response · Authors · 2025-11-28
> **Response to Reviewer PYDH**
>
> We thank the reviewer for the positive feedback about the easy-to-follow writing and constructive suggestions. Below we address the key comments raised by the reviewer.
> ## Q1&Q2: Temporal Consistency & Comparison with SOTA Video Diffusion Models
> ## A1:
> Our method does not attempt to recover full motion dynamics from a single static image. Instead, reenactment is achieved through a **compact and continuous latent motion space** with a **recurrent residual update strategy**.
>
> For the first driven frame $d_1$ and source image $s$, we estimate $\theta_{r\to d_1}$ and $\theta_{r\to s}$ via MGME and compute $\theta_{s\to d_1} = \theta_{r\to d_1} - \theta_{r\to s}$. From the second frame onward, the previously generated frame ${d_1}'$ becomes the new source ${s}'$ while the driven frame advances to frame $n$ and, so the model only needs to reconstruct the residual motion between adjacent frames $\theta_{{s}'\to d_n} = \theta_{r\to d_n} - \theta_{r\to {{d}'_{n-1}}}$.
>
> Because the model predicts only local frame-to-frame residuals rather than full motion from the original source, errors remain bounded and can be compensated by subsequent residuals, yielding stable temporal continuity except for rare jitter when a larger corrective update is needed (see **supplementary video** for visualization).
>
> We also appreciate the suggestion to compare with recent video diffusion models such as Wan2.1. These models use heavy 3D full attention modules and are trained on numerous videos, naturally achieving stronger temporal modeling capabilities. In contrast, our method is **image-based** and trained on **120h of public video**. Nonetheless, the [Visualization3](https://files.catbox.moe/fvmxkr.mp4) shows our model achieves **nearly comparable reenactment fidelity Wan2.2 (official-API)** except for mild jitter, while remaining significantly more lightweight and computationally efficient.
>
> ## Q3&Q4: The disentanglement of motion and appearance to enable cross-reenactment
> ## A2:
> **First,** although training pairs are sampled from different frames within the same identity due to the absence of cross-identity pairs, our framework explicitly learns an **identity-agnostic motion representation**. As shown in Sec. 3.2.1 and Fig. 3 of the paper, facial motion is encoded in a structured linear motion space using orthogonal codebooks. Assuming each identity has a **canonical pose (empirically verfied in Fig. 7)**, any image can transfer to it via motion features $\theta_{s\to r}$. The extracted displacement
>  $\theta_{s\to d} = \theta_{r\to d} - \theta_{r\to s}$ captures **pure pose/expression changes, independent of who the identity is**.
> This ID-free motion enables cross-identity reenactment even without cross-ID supervision.
>
> **Second,** the Multi-Group Motion Encoder (MGME) is designed to explicitly enforce this **identity-agnostic disentanglement** (Sec 3.3.1). Each motion subspace is a learnable **orthogonal codebook**, whose basis vectors represent semantic motion components independent of appearance. The **motion extractor predicts only the coefficients, preventing appearance leakage**. Motion features are then obtained via **linear combination of these orthogonal bases**, ensuring that pose changes correspond to geometry rather than identity.
>
> **Third,** While cross-reenactment performance is naturally lower than self-reenactment for all methods due to the lack of paired cross-identity training data, our model **still generalizes well across identities**. As shown in Fig. 8, WarpFace produces stable reenactment across large variations in scene, lighting, gender, and skin color, indicating that the learned motion space is largely appearance-independent.
>
> To further verify motion–appearance disentanglement, we compute the **mutual information (MI) between our motion vectors and ArcFace identity embeddings** using a MINE-based estimator. The MI remains near zero (~0.037), showing that the motion representation carries negligible identity information. This aligns with our orthogonal motion subspaces and coefficient-only prediction, confirming that motion and appearance are effectively decoupled in practice.
>
> ## Q5: Applicability to Non-Centered Face Images
>
> ## A3:
> Yes. Our method naturally handles faces that are not perfectly centered. In our training data, all videos are **center-cropped and resized to 256×256**, and the face may appear at **different spatial positions** within the crop due to natural head motion and framing variations . We also include additional [Visualization4](https://files.catbox.moe/etceps.png) to show successful reenactment on off-center faces.
>
> We hope these clarifications are helpful, and we appreciate the insightful feedback. Thanks for your time to review our paper!

---

### Official Review · Reviewer_TjYk · 2025-10-31

**Soundness:** 3
**Presentation:** 3
**Contribution:** 3
**Rating:** 6
**Confidence:** 4

**Summary:**

The paper presents WarpFace, a self-supervised diffusion framework for face reenactment that eliminates the need for domain-specific priors such as facial landmarks or 3D models. The key insight of the method is to exploit the latent motion space encoded in the diffusion process itself, improving control over facial dynamics while maintaining high-quality outputs. The paper introduces Warping-enhanced Cross-Attention (WarpCA) and Multi-Group Motion Encoder (MGME) to enable motion disentanglement. Extensive experiments show that the method outperforms several existing techniques in terms of identity preservation and image quality.

**Strengths:**

1. The paper introduces an interesting approach by using latent diffusion models to capture motion information without the need for external priors like 3D face models or facial keypoints. This is a novel way to achieve self-supervised motion learning.

2. The proposed WarpCA and MGME modules effectively address issues like semantic collapse and motion expressiveness. The geometry-aware warping in WarpCA ensures that motion features are not degraded during the denoising process, making the method robust for reenactment tasks. MGME provides better control over facial dynamics through multi-group motion representation, improving expressiveness.

**Weaknesses:**

1. While the method demonstrates solid results, the experimental improvement over existing methods like DiffusionAct and X-NEMO is relatively small in certain metrics. Some key metrics  are still worse than some established methods, which indicates that the improvement is not as dramatic as expected from a novel framework.

2. The paper mentions that SeMo and X-NEMO were reproduced by the authors. However, the authors do not provide sufficient explanation or comparison with the original reported results from the respective papers. It is unclear how the authors ensured that their reproduced results are comparable to the original.

3. It relies on iterative denoising which can hinder real-time applications. The computational cost of training and inference, especially with the multi-group motion encoding, might limit the practical deployment of the system for large-scale applications.

**Questions:**

See weakness

---

> ### Author Response · Authors · 2025-11-28
> **Response to Reviewer TjYk**
>
> We thank the reviewer for the positive feedback and constructive suggestions. Below we address the key comments raised by the reviewer.
> ## Q1: Concerns on the "small improvement"
> ## A1:
> （1）Our improvements over prior diffusion-based methods are **not marginal**.
> Across both **Self- and Cross-Reenactment**, our method improves over **DiffusionAct** by **23–62%** and over **X-NEMO** by **10–53%** across objective metrics.
> Notably, the improvements are up to ~ **62% APD/FID reduction** over DiffusionAct and **a 222% gain in User Preference** over X-NEMO(reproduced), showing the performance gains are **substantial rather than marginal**.
>
> （2）The comparison to LivePortrait is not fully aligned.
> LivePortrait relies on a **GAN coupled with a pretrained keypoint detector** and leverages **substantial high-quality internal data**, which make it even better than existing Diffusion-based methods like AniPortrait and X-Portrait[1,2]. In contrast, our model trains a motion encoder entirely **from scratch using only ~120h of public data**, offering a fundamentally different and significantly more constrained learning setting.
>
> A fairer point of comparison is DiffusionAct mentioned above. Under matched training and data conditions, our method **surpasses DiffusionAct across all metrics**, while also **eliminating the requirement for any external keypoint detector**.
> Overall, our work introduces the **first fully self-supervised diffusion framework for portrait animation** and delivers **clear compute and data efficiency with competitive or superior performance**, despite operating under considerably lighter supervision and data budgets.
> ## Q2: Clarification on Reproduction of SeMo and X-NEMO
> ## A2:
> Thanks for pointing this out. We clarify our reproduction protocol for SeMo and X-NEMO.
> Both methods rely on **large non-public training datasets and unreleased test sets**, making their originally reported numbers not directly reproducible and our reproduction can't be evaluated fairly. We followed their architectures and training settings as closely as possible, but due to the absence of their internal high-quality data, the reproduced performance is inevitably lower than the original reports.
>
> Following Review #RG3p's reminder, we evaluated the official X-NEMO implementation, which was publicly available shortly before our submission. We report a three-way comparison—**our method, our reproduced X-NEMO, and the official X-NEMO** on our public test protocol in the following table.
>
> | Method            | Self CSIM | Self APD | Self AED | Self LPIPS | Self FID | Cross CSIM | Cross APD | Cross AED |
> |-------------------|-----------|----------|----------|------------|----------|------------|-----------|-----------|
> |X-NEMO (reproduced)| 0.72      | 5.36     | 8.92     | 0.26       | 19.8     | 0.57       | 8.13      | 13.91     |
> | X-NEMO (official) | 0.77      | 4.15     | 6.24     | 0.22       | 14.9     | 0.68       | 6.34      | 9.65      |
> | Ours              | 0.79      | 2.51     | 5.58     | 0.21       | 9.6      | 0.65       | 6.10      | 10.28     |
> The results show:
> - Self-reenactment: our reproduced X-NEMO is **close to the official version**, confirming faithful reproduction.
> - Cross-reenactment: both our method and X-NEMO perform below the official model on CSIM and AED, mainly due to **training data(scale, quality, diversity)** rather than implementation differences.
>
> Thus, our evaluation is transparent and as comparable as possible given the non-public datasets of prior works.
> ## Q3: Inference Speed Limitations
> ## A3:
> We acknowledge that iterative denoising introduces slower latency for real-time applications, and we have noted this limitation in the paper. While our current implementation adopts DDIM sampling with 40–50 steps, the diffusion community has developed mature acceleration techniques—such as consistency distillation[3] that can substantially reduce sampling to **4 steps or even 1 step with only minor quality trade-offs**. These approaches are fully compatible with our framework.
>
> To clarify the computational profile, we also provide empirical measurements on a single A6000 GPU. A single denoising step takes **<0.2 s**, and the proposed **Multi-Group Motion Encoder** is lightweight and only predicts the coefficients for linear composition, adding **<0.03 s per frame**. Thus, the majority of the cost comes from standard UNet denoising rather than the motion encoding design. With established step-distillation techniques, the inference time can be improved by an order of magnitude, making the method more practical for latency-sensitive scenarios.
>
> We hope these clarifications are helpful, and we appreciate the insightful feedback.
>
> ### Reference
> [1] Aniportrait: Audio-driven synthesis of photorealistic portrait animation
>
> [2] X-portrait: Expressive portrait animation with hierarchical motion attention
>
> [3] Latent Consistency Models: Synthesizing High-Resolution Images with Few-Step Inference

---

### Official Review · Reviewer_RG3p · 2025-11-01

**Soundness:** 3
**Presentation:** 3
**Contribution:** 3
**Rating:** 6
**Confidence:** 4

**Summary:**

WarpFace introduces a self-supervised diffusion-based framework for face reenactment that requires no facial landmarks or 3D priors. Its core innovations include a geometry-aware cross-attention module (WarpCA), which predicts optical flow and occlusion maps to perform explicit spatial alignment within diffusion U-Net layers—effectively preventing semantic collapse—and a Multi-Group Motion Encoder (MGME) that decomposes facial motion into multiple orthogonal subspaces, enforcing bidirectional consistency through a symmetry constraint. WarpFace achieves superior identity preservation and visual quality over prior GAN and diffusion approaches on benchmarks such as HDTF and VFHQ.

**Strengths:**

WarpFace’s main strength lies in its elegant integration of classical geometric warping with modern diffusion-based generation, achieving a self-supervised and interpretable face reenactment framework that requires no landmarks or 3D priors. The proposed geometry-aware cross-attention (WarpCA) effectively prevents semantic collapse by aligning spatial features within diffusion U-Net layers, while the multi-group motion encoder (MGME) introduces a structured, orthogonal latent motion space with bidirectional consistency.

**Weaknesses:**

1. The overall pipeline largely mirrors traditional portrait animation frameworks: feature extraction → warping → occlusion fusion → decoding, while replacing the backbone with SD 1.5. However, the proposed method appears difficult to outperform LivePortrait baseline (Table 2).
2. Since the paper strongly claims that the method explicitly models geometric transformations, the predicted flow map F and occlusion map O should be clearly visualized to substantiate this claim.
2. It's strongly claimed that the method explicitly models geometric transformations, then flow map F and occlusion map O should be visualized.
3. If WarpCA indeed performs explicit geometric warping, applying such operations across multiple U-Net layers could risk distorting local structures and causing feature over-warping; corresponding feature visualizations are needed to verify this effect.
4. The model is not tested under extreme poses, large rotations, or heavy occlusions, where explicit geometric modeling should provide the most benefit.
5. As the official code of X-NEMO is publicly available, the authors should use the released implementation for fair and consistent comparison.

**Questions:**

please check the weakness part.

---

> ### Author Response · Authors · 2025-11-28
> **Response to Reviewer RG3p (part 1/2)**
>
> Thank you for the constructive and insightful reviews. We appreciate the recognition of our method’s strengths, particularly its self-supervised nature along with geometry-aware WarpCA alignment and overall performance. Below we address the key comments raised by the reviewers.
> ## Q1: Comparison to LivePortrait which use a similar geometry-based animation pipeline.
>
> ## A1:
> Our framework is indeed inspired by the classic portrait-animation pipeline, but the key distinction lies in the **training regime**: we adopt a **fully self-supervised diffusion formulation**, train the motion encoder **from scratch**, and use only **~120h** of public data.
>
> In contrast, **LivePortrait** relies on **(i) a GAN architecture coupled with a keypoint detector initialized from pretrained weights**, **(ii) training on substantial amounts of high-quality internal data**, and **(iii) a training pipeline that is known to be highly unstable** in our own early reproduction attempts.  The results from their original paper also shows significant improvements over several established Diffusion-based methods like AniPortrait and X-Portrait[1,2].
> For fairness, we report results using **their official released weights and inference code** rather than our reproduction, although the comparison remains **not fully aligned** due to the above differences in training setup and data scale.
>
>
> A more suitable reference is **DiffusionAct**. Using its official code and weights, **our method outperforms DiffusionAct on all metrics**—including **Self-Reenactment**, **Cross-Reenactment**, and **User Preference** settings, while removing its dependency on a keypoint detector.
>
> Overall, our method offers an early exploration of **self-supervised diffusion** for portrait animation. Its **compute- and data-efficient** design provides a clear path for future scaling.
>
> ## Q2&Q3&Q4: Visualization of Flow / Occlusion Maps and Multi-layer Warping Effects
>
> ## A2:
> We thank the reviewer for the valuable suggestion. To substantiate our claim of *explicit geometric modeling*, we have added two visualization figures in the following anoymous link: [Visualization Result 1](https://files.catbox.moe/z9qejh.png).
>
> - **Averaged FlowMap & OcclusionMap.**
>    we visualize the predicted flow map **F** and occlusion map **O** averaged over all inference timesteps. At each step, we first estimate the clean image using the predicted noise and the corresponding noised input under the current schedule, and then compute **F and O** accordingly. The results clearly reveal coherent spatial motion fields and plausible occlusion reasoning learned by our model.
>
> - **Multi-timestep visualization across U-Net refinement.**
>    To address the concern regarding potential over-warping when applying geometric warping across U-Net layers, we further visualize **F** and **O** at **t = 100, 500, 900**. As we demonstrate at Appendix D and Figure 6 that the warping based pose refinement mainly work during the middle stage of denoising, the visualization of progression also shows that WarpCA doesn't take equal effect on all steps—recovering facial structure at beginning(step 900), warping in the middle(step 500) and refining details in the end(step 100).
>
> These results support our claim that WarpCA performs **explicit and stable geometric warping** while maintaining structural fidelity.
>
> ## Q5: Experiments on extreme poses, expressions, and occlusions.
>
> ## A3:
> We appreciate the reviewer’s attention to the generalization under extreme cases. As shown in **Appendix F and Fig. 9**, our model still struggles with large rotations and heavy occlusions. This limitation primarily arises from the lack of such extreme cases in the training data, causing the learned motion representation to become out-of-distribution, which subsequently degrades generation quality.
>
> However, our method generalizes well to extreme expressions. As shown in the following anoymous link: [Visualization Result 2](https://files.catbox.moe/hm62lh.png), and supported by the quantitative results below.
> | Method                     | Self CSIM | Self AED | Cross CSIM | Cross AED |
> |---------------------------|-----------|----------|------------|-----------|
> | **Ours (original)**       |   0.79    |     5.58 |  0.65      |  10.28    |
> | **Ours (extreme expression)** | 0.78  |     5.95 |  0.63      |  11.07    |
>
> For extreme expression setting, we first compute the AED between each frame in a driving video and its corresponding source frame, and then sample one frame from the top 20% highest-AED frames (for both self- and cross-reenactment). The resulting performance remains on par with uniform sampling, indicating that our explicit geometric modeling is robust as long as the motion stays within the training distribution.

---

> > ### Author Response · Authors · 2025-11-28
> > **Response to Reviewer RG3p (part 2/2)**
> >
> > ## Q6: Comparison with X-NEMO official implementation.
> > ## A4:
> > We thank the reviewer for pointing this out. When selecting baselines, we contacted the X-NEMO authors and were informed that the **official implementation was still under internal review**, so we relied on our own reproduction. We did not re-check shortly before submission, and the **official release appeared only shortly before the deadline**.
> >
> > We have now included results using the official implementation in the following table, alongside our reproduced numbers. Our reproduction is weaker especially on cross-reenactment setting, and inspection of the inference pipeline did not reveal structural differences. The gap mainly stems from X-NEMO being trained on **substantially larger proprietary high-quality data, whereas we use ~120 h of fully open-source data.**
> >
> > | Method            | Self CSIM | Self APD | Self AED | Self LPIPS | Self FID | Cross CSIM | Cross APD | Cross AED |
> > |-------------------|-----------|----------|----------|------------|----------|------------|-----------|-----------|
> > |X-NEMO (reproduced)| 0.72      | 5.36     | 8.92     | 0.26       | 19.8     | 0.57       | 8.13      | 13.91     |
> > | X-NEMO (official) | 0.77      | 4.15     | 6.24     | 0.22       | 14.9     | 0.68       | 6.34      | 9.65      |
> > | Ours              | 0.79      | 2.51     | 5.58     | 0.21       | 9.6      | 0.65       | 6.10      | 10.28     |
> >
> > Under this constrained and reproducible setting, **our method remains strongly competitive** against the official X-NEMO results.
> >
> > Should you have another further questions, please be free to add comments. Thanks for your time to review our paper!
> >
> >
> > ### References
> > [1] Aniportrait: Audio-driven synthesis of photorealistic portrait animation
> >
> > [2] X-portrait: Expressive portrait animation with hierarchical motion attention

---

### Author Response · Authors · 2025-12-01
**Summary of Full Responses**

Across the initial reviews, all reviewers acknowledged the **novelty of our fully self-supervised diffusion framework without any human-related model and label as priors**, the explicit geometric modeling enabled by WarpCA is **elegant**(RG3p) while MGME **effectively** addresses motion expressiveness(TjYk), and the competitive empirical performance under a compute- and data-efficient setting for **further scaling potential** (PYDH).

Their main concerns centered on:
(1) comparisons with geometry-based animation method like LivePortrait (RG3p);
(2) visualization evidence supporting explicit geometric modeling (RG3p); (3) generalization under extreme poses, expressions, and occlusions (RG3p);
(4) the fairness of our X-NEMO reproduction and the need to compare with the official release (RG3p,TjYk);
(5) the magnitude of improvements over prior methods, especially DiffusionAct and X-NEMO (TjYk);
(6) inference efficiency on iterative denoising and MGME (TjYk);
(7) clarification on temporal consistency and comparison to video diffusion models (PYDH);
(8) motion–appearance disentanglement and applicability to non-centered faces (PYDH).

In our rebuttal, we provided concise yet comprehensive clarifications and new analyses addressing all raised concerns:

(1) We explained why comparisons to LivePortrait are not fully aligned, and clarified that DiffusionAct is a **more appropriate baseline** despite requiring an external keypoint detector but still inferior to our method.

(2) We demonstrated explicit geometric modeling through **averaged and multi-timestep Flow/Occlusion visualizations**, confirming stable and interpretable warping across denoising.

(3) We evaluated performance under extreme-expression settings and showed **visual robustness** as long as the motion remains within the training distribution.

(4) We included results from the official X-NEMO implementation and presented a **three-way comparison (official, reproduced, ours)**, confirming faithful reproduction and competitiveness despite data-scale disparities.

(5) We emphasized that our improvements are substantial rather than marginal, with significant gains **23–62% across metrics and over 200% in user preference**.

(6) We clarified that **MGME is lightweight and not the inference bottleneck**, while iterative denoising dominates latency, and noted that standard distillation techniques can reduce sampling to **4 and even 1 steps**.

(7) We detailed our recurrent residual motion mechanism for **temporal stability** and compared it with heavy video-diffusion models such as Wan2.2 visually.

(8) We verified motion–appearance disentanglement via **MI estimation** and demonstrated **successful reenactment on non-centered faces**.

During the discussion period, we have provided thorough and evidence-backed responses to all raised points, though we have not yet received further feedback from them. We will incorporate all clarifications, new visualizations, expanded comparisons, and updated quantitative results in the final version. With these improvements, our method is further established as the **first fully self-supervised diffusion framework for portrait reenactment**, offering explicit geometric modeling, strong cross-identity generalization under **notably smaller data and compute budgets** .

---

### Meta-Review · Area_Chair_dh2X · 2026-01-03

**Summary:**

While the paper presents a technically sound and carefully engineered system for face reenactment, reviewers raised consistent concerns regarding the impact of the contribution.  A key concern is that the paper’s methodological contribution remains incremental, combining known components, warping, occlusion handling, and motion disentanglement, within a diffusion architecture, without providing deeper theoretical insight. Empirically, while results are competitive, improvements over strong baselines are inconsistent and sometimes modest, and the benefits of explicit geometric modeling are not decisively demonstrated, particularly under challenging conditions. Additional concerns were raised regarding the breadth of evaluation, including limited comparison with recent large-scale video diffusion or foundation models and unresolved questions about scalability due to iterative denoising.

**Reviewer Concerns:**

The partially addressed concerns are listed as follows.  Added flow and occlusion visualizations help substantiate claims of explicit warping but remain largely qualitative. Inclusion of official X-NEMO results improves clarity of comparisons. Additional explanations and mutual information analysis partially clarify design intent. Inference costs is discussed more clearly.
The outstanding concerns are listed as follows.  Reviewers remain unconvinced that the method goes beyond a diffusion-based instantiation of classical warping pipelines. The rebuttal does not substantially elevate the theoretical depth. Gains over baselines are not consistently strong across metrics, and failure under extreme poses and occlusions is acknowledged but unresolved. Lack of systematic comparison with modern video diffusion or foundation models leaves the method’s relative standing unclear.

**Reviewer Scores:**

Reviewer RG3p: May maintain their marginally positive score, as many concrete technical concerns were addressed.
Reviewer TjYk: Likely to keep their score unchanged.
Reviewer PYDH: Unlikely to raise score. The core concerns about contribution level and missing comparisons remain unresolved, so the score would likely stay at reject.

---

### Decision · Program_Chairs · 2026-01-26

Reject